# Non-Renal Risk Factors for Chronic Kidney Disease in Liver Recipients with Functionally Intact Kidneys at 1 Month

**DOI:** 10.3390/jcm11144203

**Published:** 2022-07-20

**Authors:** Deok-Gie Kim, Shin Hwang, Jong Man Kim, Je Ho Ryu, Young Kyoung You, Donglak Choi, Bong-Wan Kim, Dong-Sik Kim, Yang Won Nah, Tae-Seok Kim, Jai Young Cho, Geun Hong, Jae Do Yang, Jaryung Han, Suk-Won Suh, Kwan Woo Kim, Yun Kyung Jung, Ju Ik Moon, Jun Young Lee, Sung Hwa Kim, Jae Geun Lee, Myoung Soo Kim, Kwang-Woong Lee, Dong Jin Joo

**Affiliations:** 1Department of Surgery, Yonsei University College of Medicine, Seoul 03722, Korea; mppl01@yuhs.ac (D.-G.K.); drjg1@yuhs.ac (J.G.L.); ysms91@yuhs.ac (M.S.K.); 2Department of Surgery, Asan Medical Center, University of Ulsan College of Medicine, Seoul 05505, Korea; shwang@amc.seoul.kr; 3Department of Surgery, Samsung Medical Center, Sungkyunkwan University School of Medicine, Seoul 06351, Korea; yjongman21@gmail.com; 4Department of Surgery, Pusan National University Yangsan Hospital, Pusan National University School of Medicine, Busan 49241, Korea; ryujhhim@hanmail.net; 5Department of Surgery, College of Medicine, The Catholic University of Korea, Seoul 06591, Korea; yky602@catholic.ac.kr; 6Department of Surgery, Catholic University of Daegu, Daegu 42472, Korea; dnchoi@cu.ac.kr; 7Department of Liver Transplantation and Hepatobiliary Surgery, Ajou University School of Medicine, Suwon 16499, Korea; drbwkim@ajou.ac.kr; 8Department of Surgery, Korea University College of Medicine, Seoul 02841, Korea; kimds1@korea.ac.kr; 9Department of Surgery, Ulsan University Hospital, University of Ulsan College of Medicine, Ulsan 44033, Korea; nahyw@uuh.ulsan.kr; 10Department of Surgery, Dongsan Medical Center, Keimyung University School of Medicine, Daegu 42601, Korea; gskim80094@naver.com; 11Department of Surgery, Seoul National University Bundang Hospital, Seongnam 13620, Korea; jychogs@gmail.com; 12Department of Surgery, EWHA Womans University College of Medicine, Seoul 07804, Korea; ltdrhong@gmail.com; 13Department of Surgery, Jeonbuk National University Hospital, Jeonju 54896, Korea; yjd@jbnu.ac.kr; 14Department of Surgery, Kyungpook National University Hospital, Daegu 41944, Korea; jh40356@gmail.com; 15Department of Surgery, College of Medicine, Chung-Ang University, Seoul 06974, Korea; bumboy1@hanmail.net; 16Department of Surgery, Dong-A University Hospital, Busan 49201, Korea; d002045@naver.com; 17Department of Surgery, Hanyang University, Seoul 04764, Korea; jyk1986@hotmail.com; 18Department of Surgery, Konyang University Hospital, Daejeon 35365, Korea; monjuik@kyuh.ac.kr; 19Department of Nephrology, Yonsei University Wonju College of Medicine, Wonju 26426, Korea; junyoung07@yonsei.ac.kr; 20Department of Biostatistics, Yonsei University Wonju College of Medicine, Wonju 26426, Korea; juniver1057@naver.com; 21Department of Surgery, Seoul National University College of Medicine, Seoul 03087, Korea

**Keywords:** liver transplantation, chronic kidney disease, renal dysfunction

## Abstract

Chronic kidney disease (CKD) is a critical complication of liver transplants, of which non-renal risk factors are not fully understood yet. This study aimed to reveal pre- and post-transplant risk factors for CKD (<60 mL/min/1.73 m^2^), examining liver recipients with functionally intact kidneys one month after grafting using nationwide cohort data. Baseline risk factors were analyzed with multivariable Cox regression analyses and post-transplant risk factors were investigated with the time-dependent Cox model and matched analyses of time-conditional propensity scores. Of the 2274 recipients with a one-month eGFR ≥ 60 mL/min/1.73 m^2^, 494 (22.3%) developed CKD during a mean follow-up of 36.6 ± 14.4 months. Age, female sex, lower body mass index, pre-transplant diabetes mellitus, and lower performance status emerged as baseline risk factors for CKD. Time-dependent Cox analyses revealed that recurrent hepatocellular carcinoma (HR = 1.93, 95% CI 1.06–3.53) and infection (HR = 1.44, 95% CI 1.12–1.60) were significant post-transplant risk factors for CKD. Patients who experienced one of those factors showed a significantly higher risk of subsequent CKD compared with the matched controls who lacked these features (*p* = 0.013 for recurrent hepatocellular carcinoma, and *p* = 0.003 for infection, respectively). This study clarifies pre- and post-transplant non-renal risk factors, which lead to renal impairment after LT independently from patients’ renal functional reserve.

## 1. Introduction

Because survival after liver transplantation (LT) has improved, long-term outcomes are under greater scrutiny [1]. Chronic kidney disease (CKD) is a critical long-term complication after LT that clearly affects patient survival [2,3]. Contributors to CKD in this setting are multifactorial. They include related comorbidities, perioperative renal injury, and post-transplant issues, such as the immunosuppressants used and emergent diabetes mellitus (DM) [4]. Acute perioperative kidney injury, which encompasses hepatorenal syndrome, is the foremost influence [5].

Incidence figures and risk factors for CKD in the wake of LT seem to vary according to source [6,7,8,9,10]. Such inconsistencies may reflect the heterogeneity of the reversibility of perioperative renal injury among the study population. Patients who receive irreversible renal damage are highly likely to develop permanent renal impairment during post-LT follow-up, regardless of other non-renal risk factors. On the other hand, Marit et al. have found that the five-year rate of low-level eGFR (<30 mL/min/1.73 m^2^) is only 3% in patients with no pre-transplant CKD, and the latter is rarely seen in those fully recovered from acute kidney injury post-LT [11]. Investigation with these low-risk patient subsets could provide a more precise interpretation about non-renal risk factors for CKD, which has not been studied. Furthermore, despite the existing practice-based recommendations for renal protection after LT [12], it seems that the effects of various post-transplant events on the functional decline of the kidneys in patients with no early post-transplant renal impairment are not yet fully understood.

Thus, we conducted the present investigation to identify pre- and post-transplant non-renal risk factors for CKD in liver recipients, whose kidneys are functionally intact at one month, using Korean Organ Transplantation Registry (KOTRY) data.

## 2. Material and Methods

### 2.1. Selection of Study Population

This retrospective cohort study relied on prospectively collected data of KOTRY, limited to LT-only registrants (N = 3434) between April 2014 and December 2018. Details of KOTRY have been previously reported [13]. Grounds for exclusion were the following: age < 19 years (n = 100), death or follow-up loss within 1 year after LT (n = 225), liver cancer other than hepatocellular carcinoma (HCC, n = 32), retransplantation (n = 26), eGFR < 60 mL/min/1.73 m^2^ or required dialysis at 1 month (n = 625), and lack of data (n = 212). Ultimately, there were 2214 eligible participants with intact renal function (eGFR 60 mL/min/1.73 m^2^) at the 1-month mark (Figure 1).

### 2.2. Data Collection

We retrieved registrant demographic data and medical records, including underlying primary liver disease, comorbidities (i.e., DM, hypertension), Model for End-stage Liver Disease (MELD) scores at the time of LT, and status 1 urgency ranking by KONOS (Korean Netword for Organ Sharing, described elsewhere) [14]. Hepatocellular carcinoma (HCC) status at LT (present/absent, within or exceeding Milan criteria) [15] was also obtained, as were donor characteristics (living or deceased, demographics, ABO incompatibility profiles).

KOTRY chronicles serum creatinine at the time of LT, at 1, 6, and 12 months thereafter, and then annually. eGFR may be calculated accordingly using the Modification of Diet in Renal Disease equation [16]. An eGFR of zero was applicable to patients undergoing dialysis or kidney transplantation, and we capped values at 150 mL/min/1.73 m^2^ to avoid extremes. We also categorized eGFR by grade of chronic kidney disease specified in the Kidney Disease: Improving Global Outcomes guidelines [17], regardless of demonstrable albuminuria (which KOTRY does not record in LT patients). For risk factor analysis, the eGFR at LT was intended to show the stratified risk of post-transplant CKD relative to pre-transplant renal function, whereas the eGFR at other post-transplant time points was numerically expressed. Laboratory results regarding liver function such as aspartate aminotransferase (AST), alanine aminotransferase (ALT) and total bilirubin were obtained at the same time interval. Data on the type of immunosuppressant regimen were grouped by tacrolimus plus mycophenolate mofetil, tacrolimus only, and others regardless of steroid use. Trough levels of tacrolimus were also checked in patients using the drug. Overall patient conditions 1 month after LT were gauged by the Karnofsky performance status (KPS) score, ranked as high (80–100%), intermediate (50–70%), or low (0–40%) [18].

Finally, we logged post-transplant events, such as patient death, biopsy-proven rejection, bile duct complication, vascular complication, HCC recurrence and infection (generalized infection requiring hospitalization and intravenous antibiotics). New-onset DM after transplantation was defined as the need for insulin or glucose-lowering drugs in patients without DM prior to LT.

### 2.3. Definition

CKD was defined as an eGFR < 60 mL/min/1.73 m^2^ or undergoing dialysis or kidney transplantation. Two consecutive eGFR determinations or the last follow-up value otherwise sufficed in confirming CKD. To exclude abrupt declines in eGFR due to pre-mortem multi-organ failure, the first episode of eGFR dropping below 60 mL/min/1.73 m^2^ or the need for dialysis within 3 months before patient death were not considered as CKD outcomes in a given patient.

### 2.4. Statistical Analyses

Baseline characteristics of patients who developed post-transplant CKD and those who did not were subject to comparison, using the chi-square test for categorical variables (expressed as numbers [proportions]) and the Student’s *t*-test for continuous variables (expressed as mean ± standard deviation values). 

To analyze baseline influences, we applied a multivariable Cox proportional hazard regression by entering all variables in the model. In case of significant numerical variables, the relation between the values and risk of CKD was described by the smoothing spline and visually and statistically assessed for linearity [19]. This was also performed with the multivariable method so the independent effect of each variable on CKD could be demonstrated.

Furthermore, we used the time-dependent Cox model [20] to evaluate the association between each post-transplant time-dependent factor and the risk of CKD in the following 12-month interval. Laboratory values and the type of immunosuppressant were treated as continuous variables while all the post-transplant events were treated as additive risk factors during further follow-up. To assure the relation between the post-transplant risk factors and subsequent CKD, we performed matched analyses according to the presence or absence of post-transplant events, which appeared significant in the time-dependence Cox regression. Patients exposed to risk factors were matched to controls at a 1:3 ratio based on time-conditional propensity scores [21], which were generated with baseline risk factors and eGFR treated as time-varying variables at each time point. We then examined the balance of covariates in matched populations using standardized mean differences, with values between −0.2 and 0.2 indicating adequate matching [22]. Patients outside of the balance were discarded from the matching. From the chronological matching process, patients who were selected as the control group once were not considered as potential controls in further matching, but were used again as a risk group if appropriate. Graphic depiction of the matching process is provided as Appendix A for better understanding. Subsequent CKD after the matched time points was compared by Kaplan–Meier plots and the log-rank test between those with or without specific post-transplant risk factors. All analyses were driven by standard software (SPSS v25.0 [IBM, Armonk, NY, USA] and R freeware v4.1.2 (R Foundation for Statistical Computing, Vienna, Austria), setting significance at *p* < 0.05.

### 2.5. Ethics Approval

The present study adhered to Declaration of Helsinki and Declaration of Istanbul provisions and was granted approval by the Institutional Review Board at Severance Hospital, Yonsei University Health System (IRB No. 4-2020-0915). At the time of transplantation, Korean Organ Transplantation Registry (KOTRY) obtained informed consent from all registrants. Additional informed consent was not required for this study due to its retrospective design.

## 3. Results

### 3.1. Distributions of eGFR and CKD

As shown in Figure 2, one-month eGFR values were ≥90 mL/min/1.73 m^2^ in 1201 (54.2%) and 60–89 mL/min/1.73 m^2^ in 1013 (45.8%) patients. At the time of LT, eGFRs were ≥60 mL/min/1.73 m^2^ in 1993 (90.1%) patients, whereas 36 (1.6%) had values of 15–29 mL/min/1.73 m^2^, and only 7 (0.3%) had values <15 mL/min/1.73 m^2^ (including those on dialysis). During a mean follow-up period of 36.6 ± 14.4 months, 494 (22.3%) patients developed CKD. The cumulative incidences of CKD were 14.8%, 28.9%, 31.0%, 33.4%, 37.0%, and 38.2% at 6 months and 1–5 years after LT, respectively. 

Table 1 shows a summary of the baseline characteristics for the entire population, enabling comparisons between patients with/without CKD after LT. The mean age and proportion of females were higher in patients with (vs. without) CKD. Body mass index (BMI) was also lower in patients with (vs. without) CKD. Living-donor LTs (83.5%) predominated in the population subset with intact one-month renal functionality, more so than previously reported in Korean patients [10,23]. The proportion of deceased-donor LTs and donor age were both higher in patients with (vs. without) CKD. ABO-incompatible living-donor LT was at 18.7% and proved lower in patients with CKD by comparison. Pre-transplant DM was at 23.7% overall and was higher in patients with (vs. without) CKD.

Underlying liver disease differed by CKD status. The hepatitis B virus was less frequent and alcoholic liver disease was more frequent in patients with (vs. without) CKD. KONOS status 1 designations accounted for only 1.3% and the mean MELD score was 16.0 ± 9.2, those with CKD scoring higher. Pre-transplant HCC (especially tumors within the Milan criteria) existed less often in patients with CKD. Low-level KPS scores one month after LT showed significantly greater frequency in patients with (vs. without) CKD, as did bile duct complications and infections within one month, although other surgical complications and biopsy-proven rejections were similar in the two groups. The presence (vs absence) of CKD was significantly more likely in members of the low-level eGFR subsets at LT, including 30–59 mL/min/1.73 m^2^ and <29 mL/min/1.73 m^2^. Regardless of CKD status, the mean eGFR at one month was >95 mL/min/1.73 m^2^. 

### 3.2. Baseline Risk Factors for CKD

The independent baseline risk factors for CKD (Table 2) determined by multivariable Cox modeling were as follows: age (HR = 1.21; 95% confidence interval (CI): 1.14–1.29, per five-year increment), female sex (HR = 1.43; 95% CI: 1.17–1.75), BMI (HR = 0.96; 95% CI: 0.94–0.99, per kg/m^2^ increment), pre-transplant DM (HR = 1.58; 95% CI: 1.30–1.93), low KPS score (HR = 2.01; 95% CI: 1.52–2.65, lower eGFR at LT (HR = 0.97; 95% CI: 0.97–0.99). These results are all adjusted by the numerical eGFR at one month. The relation between significant continuous variables and the risk of CKD were demonstrated in the smoothing splines (Figure 3). Age showed a positive, while BMI and eGFR at LT showed a negative linear correlation with the adjusted hazard of CKD.

### 3.3. Post-Transplant Risk Factors for CKD

The time-dependent analyses revealed that HCC recurrence (HR 1.93, 95% CI 1.06–3.53) and infection (HR 1.44, 95% CI 1.12–1.60) were independent of the post-transplant risk factors for CKD (Table 3). Other post-transplant complications, graft function examined by AST, ALT, and total bilirubin were not significant risk factors. Additionally, the type of immunosuppressant or trough level of tacrolimus were not associated with CKD in our study population. Matched analyses of time-conditional propensity scores were performed according to the presence or absence of HCC recurrence and infection with adequate balance (Appendix A). Based on Kaplan–Meier analyses (Figure 4), patients experiencing HCC recurrences (*p* = 0.013) and infections (*p* = 0.003) displayed a significantly greater propensity for subsequent CKD than the matched controls. Although AST and ALT were slightly higher in the HCC recurrence group and the infection group than the corresponding controls, as was total bilirubin in the infection group, all liver function tests were almost within normal ranges 6 to 12 months after the matched time points (Appendix A).

## 4. Discussion

This present analysis has determined pre- and post-liver transplant non-renal risk factors for CKD in those recipients with intact renal function at one month. During the multivariable Cox regression analyses, increased age, female sex, lower BMI, pre-transplant DM, and a low KPS at one month emerged as independent non-renal risk factors for CKD. The time-dependent Cox model and matched analysis indicated that the recurrence of HCC and infection after liver transplantation significantly increased the risk of subsequent CKD.

CKD after LT is largely influenced by LT-imposed acute kidney injury, including hepatorenal syndrome [24]. According to Sharma et al. [7], pre-transplant renal function is the most potent risk factor for new-onset end-stage renal disease after LT in the MELD era, where elevated serum creatinine is a major factor in deceased-donor LT prioritization. These researchers have also demonstrated that in patients receiving pre-transplant acute renal replacement therapy, the duration of such therapy, older age, and DM are risk factors for renal non-recovery [25]. However, there was no investigation into risk factors for CKD in patients without or fully recovered from pre-transplant renal impairment, which could possibly indicate that non-renal risk factors are more independent from perioperative renal damage. Consequently, we investigated pre- and post-transplant non-renal risk factors for CKD among patients with intact kidney function at one month.

In the course of the study, we corroborated well-known pre-transplant risk factors, such as age, female sex, lower BMI, pre-transplant DM, and a lower eGFR at LT. Furthermore, a remarkable finding of this study is that performance status (measured by KPS) in LT recipients with intact renal function at one month poses a significant risk of CKD. An association between physical activity and kidney function has been emphasized in several recent reports [26,27,28]; although this has not been formally documented post-LT, a considerable proportion of patients do show low performance status at the time of and shortly after transplantation [18]. This line of inquiry merits further pursuit, addressing the prospect of CKD prevention by bolstering KPS scores in liver recipients.

Another noteworthy finding of our study is that HCC recurrence is an independent risk factor for CKD, although there were several different characteristics between patients with or without HCC recurrence (Appendix A). That is perhaps explained by the nephrotoxic effects of related treatment modalities, such as transarterial chemoembolization [29], targeted therapy (e.g., sorafenib), and systemic chemotherapy [30]. In addition, repeated contrast-enhanced imaging [31] could reduce renal reserves after HCC recurrence. Although the specifics of treatment are unknown, these results are the first to confirm potential CKD risk in patients with HCC recurrences. Clinicians should be cautious with risks of CKD that interfere with therapeutic measures and result in poor survival in LT recipients who carry high burdens of HCC [32,33]. 

We also discovered that infection was a significant risk factor for CKD in our cohort. The already referenced study by Giusto et al. [6] likewise identified severe infection as an independent risk factor for renal impairment. Acute renal failure due to severe infection [34] and nephrotoxic intravenous antibiotics [35] may indeed encourage a reduction in renal function. 

The nephrotoxicity of tacrolimus is a modifiable factor for CKD after LT [36]. Achieving a reduction in or elimination of tacrolimus has been confronted in several randomized controlled trials [37,38,39]. However, the association of immunosuppressant regimens or the trough level of tacrolimus with CKD was not significant in this study. This could be attributed to the relatively lower probability of CKD in those of our study population with intact baseline kidney function. In addition, many of our population showed a reduced range of TAC trough level, as shown in Appendix A, resulting in a minimal effect on the further decline of renal function. Whether a renoprotective immunosuppressant strategy is effective in this intact kidney population needs further investigation.

There are several study limitations to acknowledge. First, we calculated eGFR values by the Modification of Diet in Renal Disease equation, which is not ideal in the context of LT [3], rather than applying the more recently developed Glomerular Filtration Rate Assessment in Liver Disease [40] or actually measuring the GFR. In addition, although we tried to exclude transient renal function declines through two consecutive CKD determinations, a fundamental drawback of KOTRY was the 6-to-12-month interval of data accrual, carrying the potential for bias when calculating time-dependent outcomes. The lack of details supporting the renal risk of post-transplant factors such as the treatment for recurrent HCC, severity of bile duct complication, and acute renal failure due to infection is another drawback of this registry study.

Using a nationwide, multicenter registry, we conducted a comprehensive analysis of pre- and post-transplant non-renal risk factors for CKD in LT recipients with intact renal function one month after grafting. Increased age, female sex, lower BMI, pre-transplant DM, and a low KPS score at one month emerged as independent baseline risk factors for CKD. Among the post-transplant exposures, HCC recurrence and infection significantly increased subsequent CKD. Our results could clarify non-renal risk factors which affect renal deterioration independently from perioperative irreversible renal damage.

## Figures and Tables

**Figure 1 jcm-11-04203-f001:**
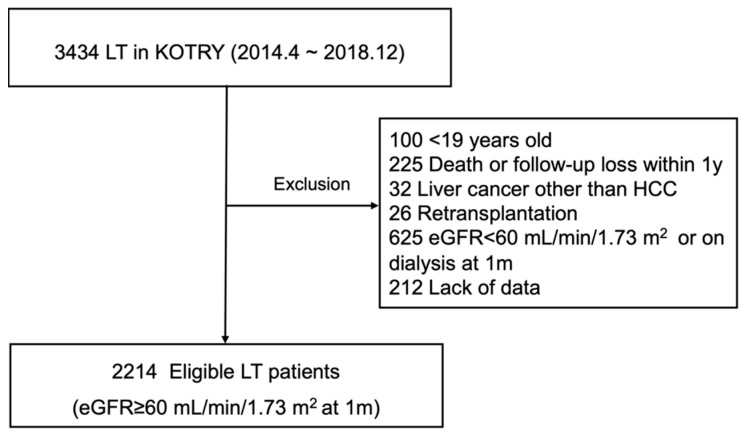
Selection of study population. eGFR, estimated glomerular filtration rate; HCC, hepatocellular carcinoma; KOTRY: Korean Organ Transplantation Registry; LT, liver transplantation.

**Figure 2 jcm-11-04203-f002:**
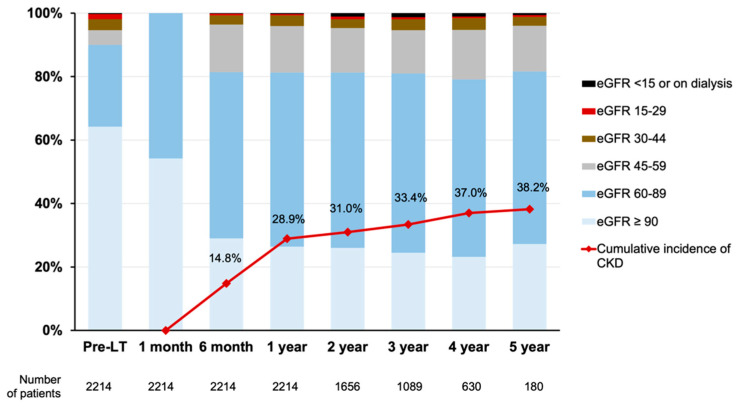
Distribution of eGFRs at specific time points and cumulative incidences of CKD. CKD was defined as eGFR < 60 mL/min/1.73 m^2^, receiving dialysis or kidney transplantation. CKD, chronic kidney disease; eGFR, estimated glomerular filtration rate; LT, liver transplantation.

**Figure 3 jcm-11-04203-f003:**
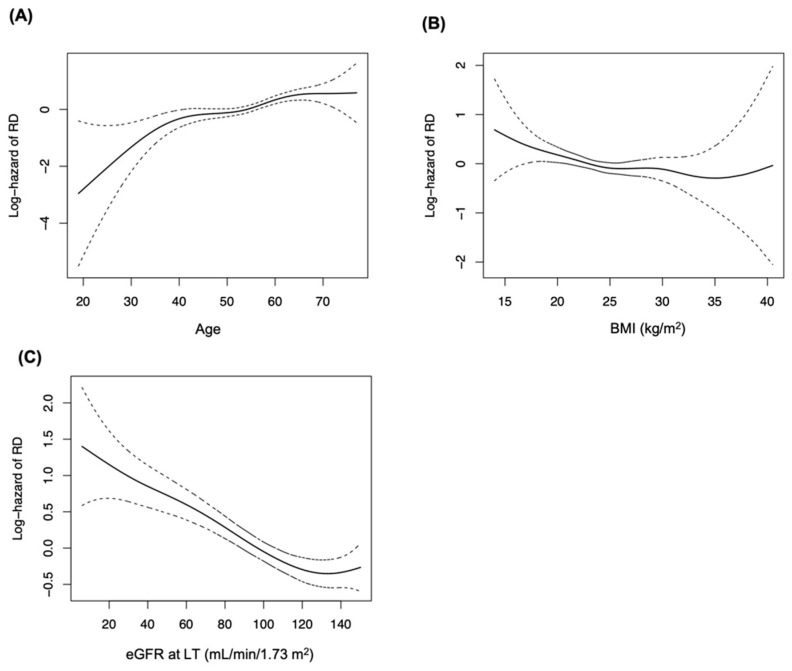
Smoothing splines showing the relation between continuous baseline variables and adjusted hazard of CKD. Splines of significant variables only were depicted, and the hazard of each variable was adjusted with all other variables. (**A**) Age, (**B**) BMI, (**C**) eGFR at LT. BMI, body mass index; CKD, chronic kidney disease.

**Figure 4 jcm-11-04203-f004:**
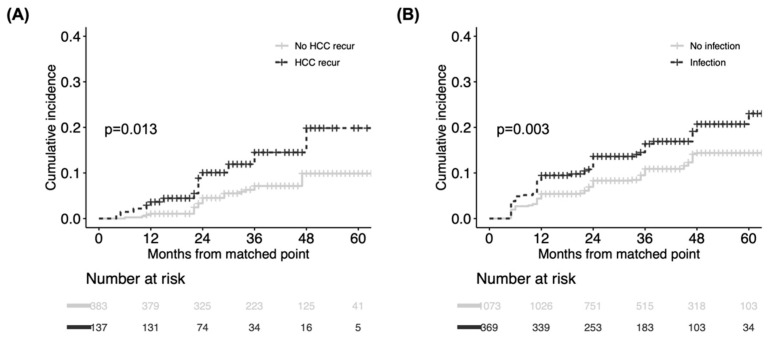
Comparison of cumulative incidence of CKD between matched groups according to presence/absence of post-transplant risk. (**A**) HCC recurrence and (**B**) infection. CKD was compared from matched time points between two groups. CKD, chronic kidney disease.

**Table 1 jcm-11-04203-t001:** Baseline characteristics of the study population.

Variables	All (n = 2214)	No CKD (n = 1720)	CKD (n = 494)	*p*
Age, years	53.4 ± 8.9	52.6 ± 9.0	56.5 ± 7.7	<0.001
Sex, female	588 (26.6)	416 (24.2)	172 (34.8)	<0.001
BMI, kg/m^2^	24.2 ± 4.2	24.2 ± 3.4	23.9 ± 3.6	0.152
Year of LT				0.013
2014–2016	1153 (52.1)	871 (50.6)	282 (57.1)	
2017–2018	1061 (47.9)	849 (49.4)	212 (42.9)	
Donor type				<0.001
Living	1848 (83.5)	1467 (85.3)	381 (77.1)	
Deceased	366 (16.5)	253 (14.7)	113 (22.9)	
Donor age, years	34.3 ± 13.4	33.9 ± 13.4	35.7 ± 13.4	0.006
Donor sex, female	799 (36.1)	627 (36.5)	172 (34.8)	0.539
ABO incompatibility	415 (18.7)	345 (20.1)	70 (14.2)	0.004
Hypertension	356 (16.1)	269 (15.6)	87 (17.6)	0.326
Pre-transplant DM	524 (23.7)	269 (15.6)	87 (17.6)	0.326
Underlying liver disease				<0.001
Hepatitis B	1265 (57.2)	1015 (59.0)	250 (50.6)	
Hepatitis C	118 (5.3)	87 (5.1)	31 (6.3)	
Alcoholic	550 (24.8)	414 (24.1)	136 (27.5)	
Cryptogenic	118 (5.3)	76 (4.4)	42 (8.5)	
Autoimmune	48 (2.2)	35 (2.0)	13 (2.6)	
Others	115 (5.2)	93 (5.4)	22 (4.5)	
KONOS Status 1	28 (1.3)	20 (1.2)	8 (1.6)	0.620
MELD	16.0 ± 9.2	15.5 ± 8.8	18.0 ± 10.3	<0.001
Pre-transplant HCC				<0.001
No HCC	1063 (48.0)	788 (45.8)	275 (55.7)	
Within-Milan	869 (39.3)	700 (40.7)	169 (34.2)	
Above-Milan	282 (12.7)	232 (13.5)	50 (10.1)	
KPS at 1 month				<0.001
High (80–100%)	823 (37.1)	665 (38.7)	158 (32.0)	
Intermediate (50–70%)	1175 (53.1)	922 (53.6)	253 (51.2)	
Low (0–40%)	216 (9.8)	133 (7.7)	83 (16.8)	
eGFR at LT, mL/min/1.73 m^2^ (categorized)				<0.001
≥90	1422 (64.3)	1221 (71.0)	201 (40.7)	
60–89	571 (25.8)	392 (22.8)	179 (36.2)	
30–59	178 (8.0)	89 (5.2)	89 (18.0)	
15–29	36 (1.6)	16 (0.9)	20 (4.0)	
<15 or on dialysis	7 (0.3)	2 (0.1)	5 (1.0)	
eGFR at 1 month, mL/min/1.73 m^2^ (numerical)	97.5 ± 25.1	101.3 ± 24.9	84.4 ± 21.4	<0.001

Data expressed as numbers (percentages) or mean ± SD values. BMI, body mass index; CKD, chronic kidney disease; DM, diabetes mellitus; HCC, hepatocellular carcinoma; KONOS, Korean Network for Organ Sharing; KPS, Karnofsky performance status; LT, liver transplantation; MELD, Model for End-stage Liver Disease.

**Table 2 jcm-11-04203-t002:** Baseline risk factors for CKD after transplantation.

	Univariable ^†^		Multivariable ^†^	
Variables	HR (95% CI)	*p*	HR (95% CI)	*p*
Age, per 5 years	1.08 (1.01–1.16)	0.033	1.21 (1.14–1.29)	<0.001
Sex, female	1.71 (1.33–2.20)	<0.001	1.43 (1.17–1.75)	<0.001
Body mass index, per 1 kg/m^2^	0.91 (0.88–0.95)	<0.001	0.96 (0.94–0.99)	<0.001
Pre-transplant DM	1.38 (1.06–1.81)	0.018	1.58 (1.30–1.93)	<0.001
KPS at 1 month				
High (80–100%)	Reference	<0.001	Reference	
Intermediate (50–70%)	2.26 (1.63–3.14)	<0.001	1.12 (0.91–1.37)	0.312
Low (0–40%)	3.25 (1.32–6.05)	<0.001	2.01 (1.52–2.65)	<0.001
eGFR at LT, mL/min/1.73 m^2^	0.97 (0.96–0.98)	<0.001	0.97 (0.96–0.98)	<0.001
eGFR at 1 month, mL/min/1.73 m^2^	0.98 (0.97–0.99)	<0.001	0.98 (0.97–0.99)	<0.001

^†^: Analyzed with uni- and multivariable Cox regression model. Multivariable model included all baseline variables. BMI, body mass index; CKD, chronic kidney disease; CI, confidence interval; DM, diabetes mellitus; HCC, hepatocellular carcinoma; KPS, Karnofsky performance status; LT, liver transplantation.

**Table 3 jcm-11-04203-t003:** Post-transplant risk factor analyses for CKD using time-dependent Cox model.

Variables	HR ^†^	95% CI	*p*
Biopsy-proven rejection	0.96	0.59–1.55	0.871
Bile duct complication	1.21	0.91–1.62	0.188
Vascular complication	0.81	0.51–1.30	0.390
HCC recurrence	1.93	1.06–3.53	0.032
Infection	1.44	1.12–1.60	0.048
NODAT ^‡^	1.17	0.81–1.69	0.420
AST, U/L	0.99	0.99–1.00	0.756
ALT, U/L	1.00	0.99–1.00	0.718
Total bilirubin, mg/dL	1.02	0.97–1.08	0.485
Immunosuppressants (with or without steroid)			
Tacrolimus + Mycophenolate mofetil	Reference		
Tacrolimus	1.34	0.65–1.41	0.745
Others	1.22	0.89–2.30	0.096
Tacrolimus trough level ^§^			
5–8 (ng/mL)	Reference		
0–5 (ng/mL)	0.97	0.77–1.21	0.857
8–12 (ng/mL)	1.04	0.81–1.36	0.701
>12 (ng/mL)	0.95	0.62–1.46	0.864
eGFR, mL/min/1.73 m^2^	0.98	0.97–0.99	<0.001

^†^ adjusted with all baseline and post-transplant factors in time-dependent Cox model. ^‡^ analyzed patients without pretransplant DM. ^§^ analyzed with patients using tacrolimus. CI, confidence interval; CKD, chronic kidney disease; DM, diabetes mellitus; HCC, hepatocellular carcinoma; LT, liver transplantation.

## Data Availability

The data that support the findings of this study are available from Korean Organ Transplantation Registry (KOTRY), but restrictions apply to the availability of these data, which were used under license for the current study, and so are not publicly available. Data are however available from the authors upon reasonable request and with permission of KOTRY.

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
