# Peer review of "Non-Renal Risk Factors for Chronic Kidney Disease in Liver Recipients with Functionally Intact Kidneys at 1 Month"

_jcm, 2022, doi:10.3390/jcm11144203_

Round 1
Reviewer 1 Report
In the present work, Deok Gie Kim et al. address the non-renal risk factors for the development of CKD beginning at stage IIIa (<60ml/min) in a fairly large representative cohort of Korean patients after liver transplantation with a large proportion of LDLT. Using adequate methodology, they conclude that infections and recurrent HCC are risk factors for the development of CKD. They further confirm the significance of age, female gender, low BMI, pre-transplant diabetes in the development of CKD after LT.
While the paper is very well written overall, I stumble on a couple aspects that need to be addressed as they fall short:
1. why are patients not included until they are 19 years old? Usually it is 18?
2. table 1. the presentation of eGFR is conceptualizing CKD. Meaningfulness?
3. tacrolimus trough levels is ng/dL? (ng/mL?) please correct.
4. it is surprising that the tacrolimus effect comes out as not significant. This is relevant for HCC recurrence. How can we better relate HCC recurrence to the development of CKD? Were the HCC patients with recurrence older, how many in particular? Had comparable immunosuppression? Induction? Rejection therapies? Perhaps the authors could elaborate on these aspects a bit more? Total exposure to tacrolimus may naturally cause concomitant renal damage and increase the risk for HCC recurrence.
Author Response
Please quote ref: Manuscript # jcm- 1818237
Thank you for the kind consideration of our manuscript, we appreciate editor and reviewers very much for their positive and constructive comments and suggestions on our manuscript:
" Non-renal risk factors for chronic kidney disease in liver recipients with functionally intact kidneys at 1 month "
Responses for the comments and questions are below.
Response to reviewer #1
- why are patients not included until they are 19 years old? Usually it is 18?
=> Maybe, the inclusion criteria "<19" was misunderstood. We included age of 19 years in this study while patients until 18 years were excluded as you pointed out.
- table 1. the presentation of eGFR is conceptualizing CKD. Meaningfulness?
=> eGFRs we presented in table 1 showed renal function before LT and 1-month baseline values of patients who later developed CKD or not. We think it is important to show baseline renal function which should be adjusted for other non-renal risk factors of CKD.
- tacrolimus trough levels is ng/dL? (ng/mL?) please correct.
=> Thank you for pointing out our mistake. We changed the units to ng/mL in table 3.
- it is surprising that the tacrolimus effect comes out as not significant. This is relevant for HCC recurrence. How can we better relate HCC recurrence to the development of CKD? Were the HCC patients with recurrence older, how many in particular? Had comparable immunosuppression? Induction? Rejection therapies? Perhaps the authors could elaborate on these aspects a bit more? Total exposure to tacrolimus may naturally cause concomitant renal damage and increase the risk for HCC recurrence.
=> Thanks for your valuable opinion. We added Table S2 showing comparison between patients with or without HCC recurrence and added next sentence in the Discussion section. Although there were differences in several variables between two groups, we can say HCC was independent risk factor for CKD in our analyses because we adjusted all available confounders in former analyses such as multivariable cox and time-dependent cox regressions.
<Page 11, line 290>
Another noteworthy finding of our study is that HCC recurrence is an independent risk factor for CKD, although there were several different characteristics between patients with or without HCC recurrence (Table S2).
In terms of TAC exposure and CKD, we didn't find any significance in our study population. We hypothesized the reason as patients include in this study had relatively good baseline renal function at early post-transplant period so that effect of TAC was relatively smaller than ordinary LT population who frequently accompanied impaired renal function. We already described it in the discussion section (page 11, line 305~)

Reviewer 2 Report
Deok Gie Kim et al. present the result of a nationwide, retrospective cohort study evaluating non-renal risk factors for chronic kidney disease in liver recipients with functionally intact kidneys at 1 month. They found that age, female sex, lower body mass index, pre-transplant diabetes mellitus, and lower performance status emerged as baseline risk factors for CKD. Moreover, they found that hepatocellular carcinoma and infection were independent risk factors for post-transplant CKD.
The overall scientific quality of the manuscript is excellent and the authors must be praised for their extensive work on the subject and a proficient level of statistical analysis. I have only a few comments regarding the manuscript.
1. CKD-EPI has proven to be a more accurate creatinine-based method of estimating eGFR in adult patients than MDRD. If possible, could the authors explain why they have chosen MDRD instead of CKD-EPI while calculating eGFR?
2. eGFR tends to change over time. Did all patients that met the CKD criteria have eGFR<60 throughout the observation period or did some of them improve? It would be beneficial to mention this in the manuscript. Cumulative CKD incidence, as shown in fig. 2 can be misleading.
3. There seems to be an error in reporting CKD status and underlying liver disease (lines 188-189). Both alcoholic disease and HBV are lower in non-CKD patients. Please re-check the interpretation of table 1.
4. Could authors specify the „infection” criteria used in the manuscript. Is it a retrospectively reported bacterial infection of any sort requiring treatment? Is it a CMV post-transplant infection? Or is it a generalized infection requiring multifactorial treatment? This needs to be addressed in the manuscript.
5. Is it possible that both lower BMI and KPS score describe a severely ill and possibly older population, with their daily activity being a reflection of their health status? Could authors speculate on that?
6. Could authors share the data or speculate on the potential effect of chemotherapy on HCC, especially in the group with recurrent HCC?
Author Response
Please quote ref: Manuscript # jcm- 1818237
Thank you for the kind consideration of our manuscript, we appreciate editor and reviewers very much for their positive and constructive comments and suggestions on our manuscript:
" Non-renal risk factors for chronic kidney disease in liver recipients with functionally intact kidneys at 1 month "
Responses for the comments and questions are below.
Response to reviewer #2
- CKD-EPI has proven to be a more accurate creatinine-based method of estimating eGFR in adult patients than MDRD. If possible, could the authors explain why they have chosen MDRD instead of CKD-EPI while calculating eGFR?
=> We chose MDRD equation based on previous literature saying both MDRD and CKD-EPI showed good performance in solid organ transplantation. Please check the next reference.
Am J Kidney Dis. 2014 Jun;63(6):1007-18. doi: 10.1053/j.ajkd.2014.01.436. Epub 2014 Apr 2.
- eGFR tends to change over time. Did all patients that met the CKD criteria have eGFR<60 throughout the observation period or did some of them improve? It would be beneficial to mention this in the manuscript. Cumulative CKD incidence, as shown in fig. 2 can be misleading.
=> As we already described in the method section (page 4, line 124), we defined CKD as two consecutive eGFR or last follow-up value were < 60 mL/min/1.73 m2 or receiving dialysis or kidney transplantation. Of 494 patients who developed CKD, 45 patients (9.1%) marked eGFR >60 at one of consecutive time points during study period. However, we are afraid that mentioning those patients in the text would confuse readers so want reviewer to agree with the current script. Please consider our definition of CKD is what has been used in many previous literatures about CKD.
- There seems to be an error in reporting CKD status and underlying liver disease (lines 188-189). Both alcoholic disease and HBV are lower in non-CKD patients. Please re-check the interpretation of table 1.
=> There was no error in interpretation of table 1 but order of components was reverse between in the table and script. We changed the sentence as below according to the order of appearance in the table.
<Page 7, line 202>
Hepatitis B virus was less frequent and alcoholic liver disease was more frequent in patients with (vs without) CKD.
- Could authors specify the „infection” criteria used in the manuscript. Is it a retrospectively reported bacterial infection of any sort requiring treatment? Is it a CMV post-transplant infection? Or is it a generalized infection requiring multifactorial treatment? This needs to be addressed in the manuscript.
=> Infection was recorded when the patient was admitted due to any infection requiring antibiotics (including antiviral and antifungal agents) in KOTRY database. Although we already described that, we added "generalized infection" in the explanation about infection as below.
<page 3, line 119>
Finally, we logged post-transplant events, such as patient death, biopsy-proven rejection, bile duct complication, vascular complication, HCC recurrence and infection (generalized infection requiring hospitalization and intravenous antibiotics).
- Is it possible that both lower BMI and KPS score describe a severely ill and possibly older population, with their daily activity being a reflection of their health status? Could authors speculate on that?
=> Thanks for your opinion. First, KPS presents physical activity and of course can reflect patient's health status. We already described possible interaction between physical activity and renal function in the discussion. Second, lower BMI could reflect severe illness, old age resulting in decreased activity, however, maybe not always. Or rather, CKD risk from lower BMI patient may derive from lower renal mass as presented in the literature (reference as below), same in older age and female sex. As those CKD risk factors have been already well-known previously, we did not added further explanation in the text. Please let us know that explanation is essential and we will revise it.
Nephrol Dial Transplant. 2009 May;24(5):1500-6. doi: 10.1093/ndt/gfn636. Epub 2008 Nov 21.
- Could authors share the data or speculate on the potential effect of chemotherapy on HCC, especially in the group with recurrent HCC?
=> As we described in the discussion including limitation section, there was no further information about treatment for recurrent HCC in KOTRY database. However, we believe that our result can suggest next topic for research about treatment of HCC and renal deterioration in LT recipients.

Round 2
Reviewer 1 Report
Dear Colleagues, I have nothing to add.
Thank you for the revision.